# Theology of Science as an Intertextual Reading: The Bible, the Book of Nature, and Narrative Paradigm

Tadeusz Sierotowicz [1,2,3]

1   Centrum Kopernika Badań Interdyscyplinarnych w Krakowie, 31-011 Kraków, Poland; sierotowicz@gmail.com
2   Istituto di Scienze Religiose di Bolzano, 39100 Bolzano, Italy
3   IISS Gandhi di Merano, 39012 Merano, Italy

**Abstract:** The paper addresses the question of the identity of theology of science, fostering its interpretation as an intertextual narration. The starting point is the consideration of the domain of theology of science, which is viewed as a third domain of truth, according to Hans Urs von Balthasar. An analysis of the Swiss theologian's perspective on this subject and the concept of God's unknowability presents a strong counterargument to the claim that the natural sciences serve as a *locus theologicus*. Theology of science, nonetheless, exists and is engaged in a lively dialogue between science and theology, encompassing both the Revelation of God and the natural world or the Bible and the Book of Nature. What kind of discourse is this? This question concerns the position of theology of science within the field of science, specifically its objectivity and rigour, according to Evandro Agazzi's analogical notion of science. Both the Bible and the Book of Nature ensure the objectivity of theology of science, while its rigour is established by the narrative paradigm. Therefore, theology of science can be seen as an intertextual narrative that engages both the Bible and the Book of Nature. The narrative paradigm of theology of science is subsequently elucidated, with particular emphasis on its cognitive aspects, narrative reasoning, the corresponding verification method, and Jewish corrective. The conclusion outlines a special task for theology of science in the modern age.

**Keywords:** domain and rigour of theology of science; *locus theologicus*; the narrative paradigm; Midrash; Hans Urs von Balthasar; Evandro Agazzi

If nature is understood as what it is meant to be for man, it can and must speak to us with the language God has given it. The question, however, is whether we do not prevent it from doing so.

(Balthasar 1967, pp. 25–26)

. . .the Lord breathes on this poor gray ember of Creation and it turns to radiance—for a moment or a year or the span of a life. And then it sinks back into itself again, and to look at it no one would know it had anything to do with fire, or light. [. . .] But the Lord is more constant and far more extravagant than it seems to imply. Wherever you turn your eyes the world can shine like transfiguration. You don't have to bring a thing to it except a little willingness to see. Only, who could have the courage to see it?

(Robinson 2004, p. 245)

The grand challenge for literature is to be capable of weaving together the various branches of knowledge, the various 'codes', into a manifold and multifaceted vision of the world.

(Calvino 1988, p. 112)

## 1. Introduction: The Identity of Theology of Science

In this paper, I propose an interpretation of theology of science as the intertextual reading of both the Bible and the Book of Nature. The proposed solution, the Bible and the Book of Nature Narrative ($B^2N^2$) Model situates theology of science within the realm of narrative paradigm, but not necessarily within narrative theology (note that the term '$B^2N^2$ Model' will be used interchangeably with the term 'narrative paradigm'). Instead, the $B^2N^2$ Model should be understood in terms of a theological narration which, as to the *genere proximum*, is near to the narrative paradigm, while the *differentia specifica* with respect to that paradigm is homologous to the *differentia specifica* that exist between the midrashic model of reading the Bible and modern theories of narration.

The paper focuses on whether theology of science is a science. To answer this question, I will use the following blueprint. I will begin with the analogical model of science proposed by Evandro Agazzi, which is briefly outlined in Section 2. This Italian philosopher teaches that the attribute of science can be credited to any rigorous reflection conducted in a certain domain of study. I will then offer a short description of the domain of theology of science (Section 3). This issue, as I will demonstrate, is strictly connected to the question of whether the natural sciences can be considered a *locus theologicus*. The discussion of that topic will introduce the pivotal section of the paper, which is dedicated to the inquiry regarding the rigour of theology of science (Section 4). I will argue that such rigour can be described with reference to the features of the $B^2N^2$ Model (Section 4.1). Subsequently, Vito Mancuso's narrative on physical evil in the universe will be briefly presented as an example of the $B^2N^2$ Model (Section 4.2). Finally, I will discuss the role of theology of science in the modern age (Section 5).

Throughout my paper, I will consider theology of science as a fully feasible enterprise, accepting the discourse of Christian-Catholic theology and its Judaic roots.[1] Following Marc Harris's suggestion, I will always write 'theology of science', dropping "the 'a' before the name" (Harris 2024, p. 37). Lastly, if there is something original in the present paper, it is the novel arrangement of ideas previously expressed by various thinkers, theologians, philosophers, and scientists (see below principles P1–P3). Therefore, I will sometimes make extensive use of nominal quotations to support my persuasive argument on behalf of different features of the $B^2N^2$ Model.

## 2. On the Analogical Concept of Science

My case for the scientific character of theology of science must begin with the question: what is science? To answer, I will adopt the point of view of Agazzi, an eminent Italian philosopher of science. There are three key words that characterize his concept of science: objectivity, rigour, and analogy (Agazzi 1979, 2014).

Agazzi's most significant contribution to the understanding of science is his original theory of scientific objectivity. He tackles the question of objectivity in both diachronic (with particular attention to the writings of Galilei) and systematic aspects. Agazzi disagrees with a view of science as merely a certain procedure without a specified domain of application. Moreover, he criticizes the naive tendency to attribute the role of the paradigm of any rigorous knowledge to science: "we now speak rather unproblematically of the sciences of economics, political science, philology, history, and so on. In other words, every field of research is today admitted as a possible branch of scientific knowledge, proviso that it is pursued in accordance with certain standards of rigour; and this implies that science is no longer characterized by what it investigates, but by how it investigates".[2]

Consequently, Agazzi devotes significant attention to the objectivity of scientific investigation.[3] His idea is simple and can be formulated in just a few words: "objectivity as reference to objects" (Agazzi 2014, p. 48). He distinguishes between objectivity in the weak and the strong sense. The former is linked to the intersubjectivity of scientific statements, while the latter expresses the fact that all scientific discourse refers to certain fields of objects. Now, in principle, two types of objects can be distinguished: existent and abstract ones (Agazzi 2014, pp. 277–93). To the former, the status of real existence is credited, while

the latter are intentional correlates of the human mind (such as mathematical objects, for example, or even the concept of God for one who does not believe in His existence). Each science, following its particular perspective, carves out, so to speak, its own domain of research in which it applies a specific method or rigour. Naturally, at least some of the characteristic traits according to which a particular science carves out its domain must be controllable or objective in a weak sense in order to ensure the intersubjectivity of its research.

To be controllable means that "an object of a certain science is simply an aspect of reality capable of being described by propositions that can be directly or indirectly assigned a truth-value by means of the criteria which are specific for that science" (Agazzi 2014, p. 89). Agazzi calls these 'criteria of protocollarity' or 'criteria of objectivity'. He intends by these expressions "those specific criteria which, within a certain science, permit the determination of which propositions are immediately true" (Agazzi 2014, p. 87).

Mario Alai synthesizes the following: "from [Agazzi's] premises derives a general and non-reductionist characterization of the concept of science: it is any form of knowledge endowed with the requirements of objectivity and rigour. Therefore, the concept of science is neither univocal nor equivocal, but analogical, that is, despite shared fundamental traits, it preserves the specificity of the various fields and styles of investigation" (Alai 2009, p. 18). Thus, the analogical concept of science can be applied in the fields of mathematics, physics, chemistry, biology, psychology, economics, and, as I shall argue, in the domain of theology of science. Agazzi's definition indicates the way to examine its application: from the identification and description of the domain to the account of the specific research style (rigour) of theology of science.

### 3. Is Theology of Science a Science? The Domain of Theology of Science

One possible answer to the question of the domain of theology of science can be sought in the first volume of Hans Urs von Balthasar's *Theo*Logic (Balthasar 2000; for more details, see Sierotowicz 2023). According to the Swiss theologian, "the world as it concretely exists is one that is always already related positively or negatively to the God of grace and supernatural revelation". Consequently, "the world, considered as an object of knowledge, is always already embedded in this supernatural sphere, and, in the same way, man's cognitive powers operate either under the positive sign of faith or under the negative sign of unbelief" (Balthasar 2000, p. 11). If the former is the case, that is, if "man's cognitive powers operate under the positive sign of faith", the truth of the world will be described "in its prevalently worldly character, without [...] ruling out the possibility that the truth we are describing in fact includes elements that are immediately of divine, supernatural provenance'. In such a case, between the domains of the natural and the supernatural, it is necessary to examine what Balthasar, following Romano Guardini, calls 'a third domain of truths, that genuinely belong to creaturely nature yet do not emerge into the light of consciousness until they are illuminated by a ray of the supernatural" (Balthasar 2000, p. 12).[4]

The third domain of truths, then, is constituted by truths 'visible' only under certain conditions: that is, only when illuminated by 'a supernatural ray' or, as an equivalent, when 'man's cognitive powers operate 'under the positive sign of faith'. Perhaps the most relevant element of this domain from Balthasar's point of view is the real distinction between essence and being (*essentia-esse*), which leads to the contingency of the world, that is, to its dependence on the Creator. As a matter of fact, if, following Balthasar, one considers 'to be' in the sense of 'to be real', then "no worldly entity can attain the coinciding of essence and reality (*essentia-esse*), even in the case of consciousness, because it can never create its own reality but must accept a reality already given to it. That is why the freest entity lives out its essence in itself but is grounded, not in itself, but in what is trans-essential, in Being absolutely".[5]

In order to more clearly define the domain of theology of science, I will recall the metaphor of the 'Book of Nature'. The expression itself "was used in Patristic times to

underline a parallelism between Holy Scripture on the one hand and the created world on the other. Since they both originated from God as their common author it was possible to regard nature as a kind of 'book'" (Pedersen 2007, p. 141). The concept of the Book of Nature embraces the universe created by God, which, as Galilei wrote in *The Assayer*, stands "continually open to our gaze" (Drake 1957, p. 237). Nevertheless, to read and understand the Book of Nature, it is necessary to learn the language "in which it is composed. It is written in the language of mathematics, and its characters are triangles, circles and others geometric figures without which it is humanly impossible to understand a single word of it; without these, one wanders about in a dark labyrinth" (Drake 1957, pp. 237–38). Strictly speaking, all of the Book of Nature can be considered the domain, as one day or another, each 'page' may be read with the help of the language of mathematics. Consequently, the domain of theology of science consists of the pages of the Book of Nature that have already been read or have yet to be read by science, illuminated by a supernatural light. Now, the latter requires a closer look.

Olaf Pedersen underlines a parallel between the two books: the Bible and the Book of Nature. With reference to Galilei's words written in his *Letter to the Duchess Christina*, he reminds readers that "the Holy Bible and the phenomena of nature proceed alike from the Divine Word, the former as the dictate of the Holy Spirit, and the latter as the executrix of God's commands" (Pedersen 2007, p. 225). This is a common Catholic doctrine. The Book of Nature can be read thanks to the natural capacities of human senses, language, and intellect, as Galilei writes in his *Letter to Benedetto Castelli*,[6] arriving at "physical conclusions of which we are already certain and sure from clear sensory experience or from necessary demonstrations" (Galilei 1613). As to the Bible, Pedersen recalls quoting Galilei. It was written with "the primary purpose of the salvation of souls and the service of God" (Pedersen 2007, p. 225).

It must be stressed that the Bible—the Holy Scripture—is the "self-revelation of God", the "revelation of the absolute reality in whose midpoint stands the figure of Christ". Consequently, theology is the second-order reflection, being, as the Revelation or the Word of God, the first-order discourse (see: Balthasar 2004, pp. 65 and 90).

In Donald Lococo's words, theology is "a systematic study of divine revelation, the speaking of God that introduces into human discourse notions and values that could not otherwise be rationally anticipated or imagined" (Lococo 2002, p. 30). It should be stressed that "theology is distinct from all other human inquiries in that its foundation is a decision to believe; it always has faith in revelation as its explicit presupposition" (Lococo 2002, p. 37, following Hans Urs von Balthasar). In other words, and giving a Catholic meaning to the sentences formulated here, the fundamental object of theological reflection is the 'content' of God's Revelation concerning Himself and the salvation of man. This 'content' of Revelation is expressed specifically in the articles of the Christian Creed, in the dogmas, in the teaching of the Magisterium, and in the testimony of Tradition, which is understood as normative. It must be added that Revelation, in terms of content, is already definitive: "The Christian dispensation, therefore, as the new and definitive covenant, will never pass away and we now await no further new public revelation before the glorious manifestation of our Lord Jesus Christ (see 1 Tim. 6:14 and Tit. 2:13)".[7]

In the case of theology of science, its domain is disclosed by a supernatural ray. It is a ray that illuminates, but it is also a ray that can be reflected. In fact, Balthasar's theological project relies on a daring presupposition. It conjectures that this supernatural light, thanks to a sort of reflection—'retrospectively cast light'—makes it possible to see peculiarities of being that illuminate such a domain (see Balthasar 2004, p. 47). With this idea in mind, Balthasar wrote his trilogy, in which theological aesthetics, dramatics, and logic are "built from within this mutually illuminating light. What one calls the properties of being that transcend every individual being (the 'transcendentals') seem to give the most fitting access to the mysteries of Christian theology. From these common properties, three were drawn out for separate emphasis: the beautiful, the good, and the true" (Balthasar 2004, p. 47). It

seems as if that reflected supernatural light, in some sense, discloses the vitality of "God's threefold personality". Balthasar writes:

> [T]ranscendence over what we think of as identical (where God is simply *causa sui*, the ground of his own self, an axiom that only produces a figment of thought) is revealed in Jesus Christ. Only in this way is God's perfect freedom unveiled as an inner vitality in which the transcendentals are identified with his identity. There is no possibility of separating the life of the three Persons from God's essence. This essence is no fourth element, something common to the three Persons. Rather, it is their eternal life itself in its processions. This is why God's 'Being' (thought of as a substance) does not manifest itself in the true-good-beautiful. On the contrary, the manifestation of the inner divine life (the processions) is as such identical with the transcendentals.
>
> (Balthasar 2004, p. 97)

Were Balthasar's supposition the case, the domain of theology of science would have become a proper *locus theologicus* by virtue of the reasoning just presented. But Balthasar, in many places, formulates a clear warning that "God cannot be 'constructed' from the point of view of the world simply by equating an infinite substantiality with the 'simple, indivisible, but non-subsisting' real" (Balthasar 2004, p. 51; see also Balthasar 2003, pp. 91–96). It is precisely this point at which Balthasar contradicts his own presuppositions. In fact, as Karen Kilby brilliantly states, "it is not difficult to find passages in which [Balthasar] specifically acknowledges the limited nature of our knowing, the need for epistemic humility, the inescapability of mystery'. Nevertheless, Balthasar 'is in fact caught in a significant performative contradiction: the way his theology is done presumes something which the content of the theology rules out" (Kilby 2012, p. 26).[8]

Therefore, the domain of theology of science—and all the more so the created world, or speaking in other terms, the Book of Nature and consequently even the natural sciences— cannot be interpreted as a *locus theologicus* without committing Balthasar's fallacy. It may seem trivial to add that, and reciprocally, theology cannot be seen as *a locus scientificus*. This separation of *loci* is but a reformulation of Galilei's comments on different purposes of both the Book of Nature and the Bible. Theology of science is neither science nor theology as such. It is a reflection that examines the natural world (the Book of Nature [BN]) as illuminated by a supernatural light (a theological light originated in the Bible [B]). Thus, strictly speaking, the domain of theology of science is constituted by these two books—both of them real and, in that sense, objective, at least for the person operating "under the positive sign of faith" (Balthasar 2000, p. 11; see also Balthasar 2004, p. 49).[9] These books, taken together, form a *locus classicus* (Barth 2010b, pp. 183–84), or, to be more precise, a *locus narrationibus* for theology of science.

How does theology of science tell its story? Or, to ask the equivalent, what is the rigour of theology of science as a science? It is the rigour of the narrative paradigm. The account of that rigour renders it possible to advance the $B^2N^2$ Model of theology of science.

## 4. Is Theology of Science a Science? The Rigour of Theology of Science

The rigour of theology of science is determined by its intertextual nature. It is necessary to clarify that throughout this essay, the term 'intertextual' is used in its letteral sense, that is, to indicate the narrative connection between different texts. Therefore, it has none of the post-structuralist or postmodern meaning given to the term by the thinkers collaborating with the *Tel Quel* or *L'Infini* journals, like Julia Kristeva, Philippe Sollers, Roland Barthes, or Gerard Genette.

Intertextuality, as intended here, signifies an overarching narrative space open between the Bible and the Book of Nature,[10] a sort of book with blank pages on which a theologian of science can write down a diachronically or synchronically knitted narrative plot that presents an account of events and responds to the specific quest for their meanings, thus giving a more profound intelligibility, understanding, and participation to knowledge

about God, the world, and ourselves.[11] The presence of the postulated, blank book between the two books involved in the narration signifies that theology of science strives to account for the gap of meaning and tell a coherent story. What is at stake, however, is a gap of meaning, not a causation gap. Hence, theology of science does seek the exact point where God's 'finger' enters into contact with and moves the physical world, whether it be a point somewhere in the subatomic world or a point somewhere in the midst of neurons in the human brain. To reiterate, then, a narrative in the context of theology of science is engaged in creative activity that allows it to fill, in a meaningful and coherent way, the gap of meaning between the Bible and the Book of Nature.

It is vital to emphasise that narrative, as understood here, not only conveys meaning but also, contextually, provides a compelling or good reason to act and make choices. Theology of science is a form of narration, but it has yet to be seen as a branch of narrative theology. In fact, theology of science, which is both theological and scientific narration, applies the features of the narrative paradigm rather than the rigours of this or that model of narrative theology.[12]

The following propositions specify the rigour of the narrative paradigm or, in other words, outline key aspects of the $B^2N^2$ Model of theology of science:

**(P1)** The $B^2N^2$ Model has significant cognitive aspects ('logos' and 'mythos', 'Gestalt' and mental models).

**(P2)** The $B^2N^2$ Model is based on narrative reasoning and the corresponding mode of verification (narrative probability and narrative fidelity, believability and consilience, illative sense).

**(P3)** To create theology of science narration, both theological and scientific narration (and not a solely scientific or solely theological one), the Jewish corrective is required.

The next section will elaborate on these propositions to describe a refined structure of the $B^2N^2$ Model.

### 4.1. Key features of The $B^2N^2$ Model

### 4.1.1. P1: The $B^2N^2$ Model and Its Cognitive Aspects

The abbreviation '$B^2N^2$ Model' consists of three components: $\mathbf{B^2N}$ for the Bible and the Book of Nature, $\mathbf{N}$ for Narration, and lastly, the term **Model**. I will start with narration as seen in the context of the narrative paradigm.

**(N)**. A systematic description of the narrative paradigm was proposed by Walter Fisher several years ago as a communication paradigm (see Fisher 1984, 1985; Stroud 2016). "To begin, narrative is generally the representation of an event or sequence of events. It is closely tied to description, but it goes beyond description" (Worth 2008, p. 43). Where theology of science is concerned, the narrative goes beyond the description of a given event, sequence of events, or change in the states of affairs taken from the Bible and the Book of Nature. Theology of science instead seeks the understanding that can be gained from engaging with narrative. As for the term 'paradigm', I adopt the meaning attributed to it by Fisher, slightly changed and adapted to the perspective of theology of science. A paradigm, then, is a representation designed to formalize the structure of two components: scientific achievements, that is, deciphered pages of the Book of Nature, and theological beliefs (rooted in the Bible), with the aim of arriving at a more profound understanding and exploring the what is associated with the topic of the narration. The narrative paradigm, in turn, "can be considered a dialectical synthesis of two traditional strands in the history of rhetoric: the argumentative, persuasive theme and the literary, aesthetic theme" (Fisher 1984, p. 2). I will elaborate on an argumentative aspect of the $B^2N^2$ Model later (see P2 below). Next, I will focus on the persuasive aspect of the narrative paradigm.

Fisher's narrative paradigm reunites *logos* and *mythos* in the sense Fisher attributes to these terms. Fisher, writing about Kenneth Burke and his contribution to the modern understanding of rhetoric, recalls Burke's affirmation that "Wherever there is persuasion, there is rhetoric and wherever there is meaning, there is persuasion". For Burke, Fisher

comments, rhetoric "is not purely an epistemological transaction', and it 'works by identification rather than demonstration". In that way, "reason as well as aesthetic qualities" are recaptured "in all forms of human communication, [which] reinforms the original sense of *logos*" (Fisher 1985, p. 86). Fisher uses in his comment the term 'reinforms' because "in the beginning was the word or, more accurately, the *logos.* And in the beginning, *logos* meant story, reason, rationale, conception, discourse, and/or thought". It was only later, after Plato and Aristotle, that the terms *logos* and *mythos,* firstly conjoined, were separated. As a consequence, "*logos* was transformed from a generic term into a specific one, applying only to philosophical (later technical) discourse; poetical and rhetorical discourse were relegated to a secondary or negative status in regard to truth, knowledge, and reality" (Fisher 1985, p. 74).

What is a *mythos* for Fisher? He gives an answer in the context of human communication: "some persons know more than others, are wiser, and are more to be heeded than others. But no one knows all there is to know, even about his or her own area of specialization. Human communication in all of its forms is imbued with *mythos*—ideas that cannot be verified or proved in any absolute way, including metaphor, values, gesture, and so on—as well as, on occasion, clear-cut inferential or implicative structures" (Fisher 1985, p. 87). In Fisher's opinion, *mythos* have relevant cognitive significance and are intrinsic to the narrative paradigm.

**(B$^2$N)**. The abbreviation refers to the Bible and the Book of Nature, and it signifies that there are two different plots interwoven in one narration. This circumstance is not new in the theory of narrative, as Peter J. Rabinowitz notes in the context of his analysis of detective stories written by Dostoyevsky and Faulkner. He emphasizes that there are two plots in these stories: on the one hand, there is a detective story, in which careful investigation (*query*) leads to the identification of 'whodunnit'. On the other hand, there is a plot of a philosophical nature, in some ways more complex, which leads to a greater understanding of the man's situation in the reality around him. Rabinowitz attributes to these stories the status of 'discovery novel':

> [T]he novel of discovery is structured in such a way as to move towards a climax in which the protagonist suddenly and usually unexpectedly acquires key knowledge, experiencing a sudden enlightenment of the "meaning of things" that radically changes his worldview. This flash of insight [...] usually involves ethical, metaphysical, or social rather than scientific knowledge; and it usually forces the protagonist to re-examine his previous relations to the world.
>
> (Rabinowitz 1979, p. 356)

Following this description, the narrative of theology of science can be attributed (via analogy) to the same status. As a matter of fact, in each theology of science narration, there are two plots: on the one hand, a scientific story, which through a careful query, investigates this or that page of the Book of Nature, and on the other hand, a theological plot of a different character. These two plots are then knitted into one narration, which leads to the discovery of the meaning and a greater understanding of the reality that surrounds man.

What happens in a discovery novel is a sort of consilience of two plots, which leads to a new *Gestalt*—in this case, to a new organization of two different plots taken from the Bible and from the Book of Nature. This *Gestalt* belongs to the 'universe of discourse' of theology of science, to use Agazzi's terms. Agazzi notes that in the literature on the subject, the 'universe of discourse of a science' means 'the domain of the individuals of that science'. This notion involves two assumptions: the first assumption is that by 'universe of discourse', we must understand a set of entities endowed with properties and relations about which a given discipline or theory is supposed to speak. The second is the identification of meaning with reference. In fact, when one speaks of a 'domain of discourse', one is using an expression that is in itself linguistic in nature. Simply put, these terms refer to the framework in which the discourse is intended to be meaningful, as Agazzi explains. In this sense, the universe of discourse can be called a conceptual space characteristic of each science and, within a science, of its various theories. The structure or

configuration of this conceptual space could then be seen as a new form (*Gestalt*) in which several already known details are organized in a different way or suddenly appear to be relevant to each other in a way not previously recognized (see: Agazzi 2014, pp. 139–40). The emergence of a new *Gestalt* gives insight, or 'produces' meaning, a specific narration that theology of science is seeking.[13]

What is the meaning of the verb 'produce' in this context? The 'machinery of production' is put in motion whenever something new is created. This applies to mathematics, physics, philosophy, music, theology, and literature.[14] Readers of Agatha Christie will remember that in *Murder on the Orient Express*, Hercules Poirot solves the case by coherently ordering the framework of clues around the frequently recurring number twelve: there were twelve passengers in the Istanbul-Calais carriage, the victim was stabbed twelve times, a jury is composed of twelve people (Christie 2011, p. 215). There is no space here to thoroughly describe the stages of the creative process, or as Poirot says, of his 'scheme of 'guessing' (Christie 2011, p. 211) that led him to the solution. However, it should be noted that the emergence of a new organization, of a new understanding of facts, seems to occur around centres of crystallization (i.e., around focal points of interpretation, like the number twelve in Poirot's case). Probably the most apt description of the 'scheme of guessing' mentioned by Poirot is offered by the theory of abduction, that is, 'the inference that makes science' (McMullin 1992).

**(Model)**. The term 'model' refers to a 'mental model'. Models can be considered alongside 'thought experiments', which constitute a fundamental evolutionary achievement of humankind.[15] The notion of mental models dates to ideas formulated by Charles Peirce and Ludwig Wittgenstein. Often, the presentation begins with Kenneth Craik's theory, which identified the ability to predict events as a fundamental property of human thought and a particularly advantageous adaptive achievement. Craik described mental models with a strong emphasis on their ability to mirror the processes of the real world, both in terms of their structure and the processes that occur there:

> [B]y a model we [...] mean any physical or chemical system which has a similar relation-structure to that of the process it imitates. By 'relation-structure' I do not mean some obscure non-physical entity which attends the model, but the fact that it is a physical working model which works in the same way as the process it parallels, in the aspects under consideration at any moment.
>
> (Craik 1943, p. 51)

In short, mental models have a structure similar to the structure of a 'fragment' (in the case of the present discussion, fragments) of the reality they represent. According to Craik, the construction of mental models belongs to the natural capacities of the human mind. The mental process that constitutes a model is based on three fundamental processes: translation of the processes of the external world into words, numbers, or symbols, reasoning (i.e., the transition to other symbols through deduction, induction, etc.), and finally, re-translation of these symbols into external processes or the recognition of the correspondence between symbols and external processes.

Since the 1980s, mental models have been studied intensively by philosophers, theologians, literary critics, computer scientists, and cognitive scientists. According to scholars who follow Craik's approach, mental models are constructs of the human mind that represent situations, events, and processes for solving certain problems. These models can be dynamically manipulated, transformed, and developed through forms of reasoning that differ from the forms of reasoning applied to systems of propositions, such as deduction or induction. A recourse to thought experiments is often cited as one example of this specific type of reasoning.[16] Ultimately, it is relevant to point out that some scholars interpret narrative in terms of mental models and thought experiments (see Nersessian 1993, pp. 293–95; Elgin 2017, pp. 221–47; Fehige 2019; Stierstorfer 2022). To sum up, theology of science functions like a discovery novel in that its narration can be interpreted as a mental model of the question which represents the narration's subject matter.

### 4.1.2. P2: A Narrative Reasoning and Verification of the $B^2N^2$ Model

The 'flash of insight' mentioned above with reference to the discovery novel (which emerges thanks to the presence of a centre of crystallization) enables the two-plot narrative process, which results in a *Gestalt* and a related mental model. Here is the proper place of the *mythos* component of the narrative paradigm. It appears as an idea that originates a new way of understanding and gives the reader/hearer courage to live in the world. As Aldous Huxley writes in reference to the words of Clémenceau, "Ce qui donne du courage, ce sont les idées".[17] But the $B^2N^2$ Model is not only *mythos*. It is also *logos* in Fisher's meaning of these terms. The next section will scrutinize the latter: the reasoning which is specific to the narrative paradigm.

It has already been noted that narratives organize how the world is perceived, as well as respective forms of knowledge. This procedure, which can rightly be called 'narrative reasoning', leads to verisimilitude and is distinct from discursive reasoning, which is built on logical forms of argumentation and scientific procedures. In this framework, Fisher proposed the concept of the *homo narrans*, whose sense of self is formed through the narration of his own history, identity, and place in the world, among others. *Homo narrans'* way of reasoning is determined by his inherent awareness of narrative probability and constant habit of testing narrative fidelity. He evaluates what constitutes a coherent story and ensures that the stories he experiences sound true to the stories he knows to be true in his life (see Fisher 1984, pp. 7–8).

In the words of Jerome Bruner, "we organize our experience and our memory of human happenings mainly in the form of narrative-stories, excuses, myths, reasons for doing and not doing, and so on".[18] Narrative, as such, is a conventional form transmitted culturally. Unlike the constructions created by logical and scientific procedures that can be falsified, narrative constructions can be attributed to some degree of verisimilitude. Narratives, then, are a version of reality whose acceptability is governed by convention and narrative necessity rather than by empirical verification and logical necessity.

Narrative reasoning is descriptive in its character. It offers an understanding of human beings and their actions and is not the 'laws of thought' that can be taught. Instead, the traditional reasoning, that of *logos* in Fisher's sense, *can* be taught. It posits a set of rules of prescribed calculation or inference-making. In that sense, *logos* has a normative character. But what makes theology of science narration a credible story? Or, asked another way: what makes one given story better than another?

As previously stated, narrative probability and narrative fidelity appear as two features of narrative reasoning.[19] The former involves a coherence theory of truth, with the consistency of narration with the tenets of the Bible and the Book of Nature at stake. The latter involves "how people come to adhere to particular stories" because they are seen as "more sound than another". In that sense, the narrative paradigm provides 'good reasons' to accept the story. Fisher suggests that "the logic of good reasons is the most viable scheme presently available by which narratives can be tested. Its application requires an examination of reasoning and "inspection of facts, values, self, and society". In epistemological terms, "narrative fidelity is a matter of truth according to the doctrine of correspondence" (Fisher 1984, pp. 15–16). In that case, a sufficiently convincing correspondence of the $B^2N^2$ Model to the world in which a storyteller lives provides plausible reasons for belief and action.

The correspondence mentioned above evokes the classical definition of truth. As the present discussion of the $B^2N^2$ Model is not supported by a formal apparatus, it is not possible (nor, in my opinion, necessary) to develop a strictly technical formulation of this affirmation. Nevertheless, if we accept Agazzi's assertion that scientific theories "are essentially sets of sentences connected" in case of theology of science by narrative links, a theory is essentially "a unique sentence of considerable length" (Agazzi 2014, p. 387). If so, it is possible to say whether a given theory "can be more or less 'faithful', 'accurate', or adequate to the description of the domain of objects [it tries] to interpret in making a

given *Gestalt* explicit, and this notion of 'adequacy' is precisely the one used in the classical definition of truth" (Agazzi 2014, p. 388).

Narrative fidelity also posits the question of verification of the persuasive ability of reasoning. While logico-scientific reasoning, or as Sarah Worth puts it, discursive reasoning, persuades because of the strength of the laws of thought applied, of their conclusions, and possibly their truth, narrative reasoning convinces because of its believability.[20] In consequence, "narrative lines of reasoning do not generally prove anything, but they do show how something might have come to be the case" (Worth 2008, p. 49). The fact that narrative lines of reasoning do not prove anything beyond any doubt promotes a worldview that "is not a set of puzzles to be solved. Instead, the world is known as a story, and there are always a range of potential stories to choose among in explaining the world and our place in it" (Stroud 2016, p. 2).

The concept of narrative believability can be further explained in reference to consilience and illative sense. These two notions are helpful in exploring why people accept this or that narration. Consilience is William Whewell's term, but my use of it is somewhat different. Whewell coined several neologisms, including 'the colligation of facts', 'the explication of conceptions', 'the decomposition of facts', or 'the superinduction of conceptions'. In his writing on the philosophy of the author of *The Philosophy of the Inductive Sciences*, Larry Laudan emphasizes that the concept of 'the consilience of inductions' is one of the most productive and important doctrines in Whewell's methodology. Laudan writes that consilience is "not only vital to a comprehension of his philosophy of science, but it also the key to his historiography of science, for it is largely in terms of consiliences that Whewell formulates his theory of the *progressive* nature of scientific growth and evolution" (Laudan 1971, p. 368; Laudan's italics). Richard Di Rocco applies this concept to theology and science discourse, accepting Edward Wilson definition: consilience means "literally a 'jumping together' of knowledge by the linking of facts and fact-based theory across disciplines to create a common groundwork of explanation". But here Di Rocco's approach diverges from mine. For Di Rocco, as well as for Wilson, consilience "takes place when an Induction, obtained from one class of facts, coincides with an Induction obtained from another different class', confirming 'the truth of the Theory in which it occurs" (Di Rocco 2018, p. 116). Instead, I use the term 'consilience' in a much weaker sense. Given the different rigour of theology and the natural sciences, the consilience means a coherent convergence and does not confirm the truth of theology of science narration in which 'inductions obtained' coherently converge but only contributes to the discovery of meaning, reinforcing the believability of a given narration.

The mention of convergence leads to the concept of the illative sense proposed by John Henry Newman.[21] The English cardinal addresses the subject in some of his writings, mainly in *The Grammar of Assent*. The illative sense is about how human beings arrive at certitude. This common thought process is nothing more than "the sole and final judgment on the validity of an inference in concrete matter [. . .] committed to the personal action of the ratiocinative faculty, the perfection or virtue". Newman sometimes compares this faculty or instinct to a 'good sense', a 'common sense', or a 'sense of beauty' (see Newman 1874, p. 345). The illative sense represents a capacity to judge and draw conclusions about a totality to which different, individual propositions lead through a convergence of clues. It must be emphasised that for Newman, the illative sense is a faculty of the mind: "it is the mind that reasons, and that controls its own reasonings, not any technical apparatus of words and propositions" (Newman 1874, p. 343).

Although the illative sense is a faculty of reasoning, one cannot say that its resulting inferences have a formal character in the sense of the lines of reasoning proper to classical logic. Consequently, "the illative sense [. . .] supplies no common measure between mind and mind, as being nothing else than a personal gift or acquisition" (Newman 1874, p. 362). This is why the certainty provided by inferences of this kind remains similar to what is usually called a 'moral certitude'. In other words, for Newman, a proposition is certain (*certainty*) if the human being himself is certain in the sense of subjective certainty (*certi-*

*tude*). Similarly, "certitude is a mental state", while "certainty is a quality of propositions" (Newman 1874, p. 344).

In Newman's words, "the formal logical sequence is not in fact the method by which we are enabled to become certain of what is concrete [. . .]. It is the cumulation of probabilities, independent of each other, arising out of the nature and circumstances of the particular case which is under review" that leads to certitude of a given narration (Newman 1874, p. 288). The point is, as Newman writes, that the rule of acceptation "for one man is not always the rule for another, though the rule is always one and the same in the abstract, and in its principle and scope" (Newman 1874, p. 356). Hence, with the Bible and the Book of Nature taken together, many stories can be told. Nevertheless, the question arises: are there any limits to this storytelling? For a possible answer, I investigate what is called the 'Jewish corrective'.

### 4.1.3. P3: The Jewish Corrective of the $B^2N^2$ Model

The expression 'Jewish corrective' was used by Jean Baptiste Metz, who rightly considered "storytelling [. . .] a specifically Jewish strength" (Mauz 2009, p. 266). Perhaps the most evocative example of this storytelling strength is Midrash. Midrash is a specifically Jewish mode of exegesis of the biblical text that was developed by rabbis writing between 400 and 1200 AD.[22] The term refers to a way of interpreting biblical text, as well as to a literary genre that led to an extremely rich corpus of works. In this subsection, I will focus on Midrash understanding as a way of reading biblical text that combines exploration and the posing of critical questions.[23]

How does Midrash proceed? Let me use an example. Genesis 22 tells the well-known story of *Akedah*, short for the 'binding of Isaac'. God calls Abraham and commands him to take his son Isaac and offer him as a sacrifice on Mt. Moriah. In the end, having seen that Abraham fears Him, God sends a ram so Abraham can offer it instead of his young son. Astonishingly, immediately after the sacrifice, the name of Isaac disappears from the pages of the Bible until Chapter 24, when the story of his wedding is told. The Scripture is silent on whether Issac was angry with his father, whether he understood what had happened, or whether Abraham and Isaac ever conversed, exchanged ideas, or even, perhaps, argued after such a dramatic moment (see Wiesel 1976, pp. 69–97).

The Midrash, which attempts to fill this textual and narrative gap, proposes a kind of extended account in light of the changed historical and cultural situation. It does so without changing the biblical text, as far as the words are concerned. This approach, characterized by an unconditional acceptance of the biblical text, opens the hearts and minds of the listener to the values, issues, and theology that underlie the biblical story. In doing so, Midrash freely crosses the boundaries of literary genres by combining the biblical parable with other forms of narrative (Stern 1998, p. 31). By its very nature, Midrash is an interdisciplinary endeavour of telling the story 'between the words' of the Text.

Ithamar Gruenwald proposed a generalisation of the midrashic approach to the Text, introducing the concept of 'the midrashic condition'. He describes the condition by stating that "scholars and literary critics have gradually realized that Midrash as a literary genre and form of interpretative expression is present in almost all form of literary creation" (Gruenwald 1993, p. 6). Gruenwald then underlines the interpretative elasticity and creativity of Midrash while simultaneously conveying its role in preserving Jewish religious and cultural tradition. Furthermore, Gruenwald comments on the cognitive potential of Midrash as "a vital instrument in creating patterns of perception, conceptualization, and realization in which scriptural terms of reference are applied for existential. [. . .] Midrashic-like modes of relating to a scriptural or canonical text can be extended to any type of mental relationship that entails the concern for establishing relevance or relatedness to any given fact or piece of information" (Gruenwald 1993, p. 7). For Gruenwald, Midrash is not a mere strategy for understanding texts. In fact, "Midrash not only creates exegetical information, [. . .] but also the spheres of meaning" (Gruenwald 1993, p. 9). By and large, "Midrash

should be viewed as a mode of cognition in the same sense as literature, philosophy, or science" (Gruenwald 1993, p. 13).

Midrash's source of fruitfulness, and at the same time, its difficulty (at least for scholars not engaged in the Jewish tradition) is the polysemic nature of Midrash. In short, as Myrna Solotorevsky notes, "each element of the Bible (letters, words, verses, chapters) is allowed to function as an autonomous unit which has endless possibilities of combination with other units", and thus "provokes the polysemic radiation of the text". In that sense, "Midrash is the metatext of a sacred text", but—as Solotorevsky highlights—"it must apparently revolve around a centre (the Scripture as the Word of God)" (Solotorevsky 1986, pp. 254–55).

Solotorevsky wrote her paper beginning in the 1980s, during a period when scholars in English and Jewish studies aimed to connect Midrash with literary theory criticism. This 'Midrash–theory connection', as it was later called, produced a stream of papers and important monographs, but its influence weakened by the end of the 1990s (see Freer 2016, p. 336; for an extensive overview with commentary, see Catlin 2023). David Stern described the reason for the decline of this extraordinary interdisciplinary research in his paper on Midrash and indeterminacy. Stern's answer indicates the precise point where the Jewish corrective comes in.

Midrash, because of its polysemy, frequently leads to contradictory opinions from rabbis of different schools. However, for rabbis, on the condition that the biblical text remains unchanged, different opinions do not exclude each other. What is important is the conviction that the sanction "for such paradoxical truth [...] is the common divine origin" of different interpretations. "This divine sanction for Scriptural polysemy also differentiates the midrashic concept of polysemy from its post-structuralist counterpart, indeterminacy". However, it is not the style that differentiates Midrash from postmodern indeterminacy, "but rather the latter's formal resistance to closure" caused by the final revelation of only one perspective. Postmodern indeterminacy predicates "the absence of one and only one context from which to view the flux of time or the empirical world, of one and only one method that would destabilize all but itself, of one and only one language to rule understanding and prevent misunderstanding". Midrashic polysemy, on the contrary, "is predicated precisely on the existence of such a perspective, the divine presence from which all the contradictory interpretations derive". The polysemy of Midrash, referred in the last instance to the divine origin of Torah is, as Stern notes, "a claim to textual stability rather than its opposite". For that reason, as Stern comments, whereas "the ruling interpretive ideologies of Western culture [...] may be said to be motivated by an anxiety over the loss of meaning or presence [...], Rabbinic interpretation is not worried by the possible absence of meaning, by a fear that presence in the text may be irrecoverable or lost" (all quotes in the present paragraph are from Stern 1988, pp. 153–54).[24]

What does Midrash mean for theology of science? As reported above, Myrna Solotorevsky writes that Midrash revolves around a centre, which is the Bible. The narration of theology of science revolves around two centres, two *heteronomous* utterances of the same author, looking for meaningful coherence between them (see Mauz 2009, p. 278). The presence of two centres of rotation is the greatest challenge for theology of science, especially when examined through the $B^2N^2$ Model. Here, on the one hand, is the Bible, a text characterized by polysemy of words, leading to multiple interpretations of the same unquestionable Text. On the other hand, the Book of Nature, as written in the language of mathematics, fosters the univocity of its reading expressed in mathematical theories confirmed by experiments.

The revolution around two centres proceeds following two laws of rotation. The first law concerns the distance and separation between two centres; the second describes the trajectory of the rotation. As to the first—Menachem Kellner, in writing about science and understanding of Torah (and hence about what, in some sense, corresponds to the dialogue between science and theology[25]), invokes a sort of principle of modesty. This idea, in fact, rephrases Galilei's statements from his letter to the *Letter to the Duchess Christina*. Kellner writes:

to reject the claim that the earth is vastly old, for example, is not only to reject the science of geology, but the entire edifice of contemporary physics and chemistry. The cosmos simply cannot be under 6000 years old. This, of course, is only a problem for the most stubborn of biblical literalists. [...] In these and other matters, the Written Torah cannot be taken literally without rejecting the crushingly overwhelming weight of scientific evidence. [...] On God's existence, the creation of the cosmos, Sinaitic revelation, providence, prophecy, miracles, efficacy of prayer, the special relationship of God to the Jewish people, divine reward and punishment, etc., science seems to have little definite to say to us, and it appears to me, is not likely to have much to say in the foreseeable future. [...] The Torah cannot contradict that which has been *proven* scientifically but science often proves less than what some scientists think they have proven. We must live in a world of fewer absolutes than many thinkers (rabbis and scientists alike) would like.

<div align="right">(Kellner 2020, p. 10; Kellner's italics)</div>

Strictly speaking, Kellner's version of the principle of modesty draws impassable silhouettes around each of the two centres: silhouettes that can change with time but which maintain two centres that are distinct and impenetrable to each other. However, the narration of theology of science attempts to give an account of a specific query that engages both books. It is, as previously stated, the complex rotation around two centres—around midrashic polysemy and scientific univocity—which leads to the question of the second law, the law of rotation around these centres. The movement under discussion cannot pierce the silhouettes mentioned above. The revolution must necessarily embrace two centres. Otherwise, one would only have to deal with either the scientific or the theological narration. For that reason, theology of science narration should not change even a 'single jot' of the Bible (Matthew 5, 18) or of the Book of Nature. Midrash refreshes, reaffirms, and interprets the biblical text and, at the same time, fully respects it (Taggart 2010, p. 18). Likewise, the narration of theology of science refreshes, renews, and interprets both biblical and scientific texts while simultaneously respecting them. This is precisely the Jewish (midrashic-like) corrective. It confirms both the theological and scientific character, as well as objectivity in Agazzi's sense and non-fictional features in Fehige's meaning of the narration at hand.[26]

### 4.2. The $B^2N^2$ Model and the Discovery of Sense—An Example

Vito Mancuso proposed a helpful example of the $B^2N^2$ Model in his book on the problem of physical evil in the universe created by God and specifically on the condition of handicapped persons. Mancuso writes that this topic is particularly important to him for personal reasons.[27] He draws a strict connection between the issue of the suffering of the innocent and the freedom of human beings. In his book *Dolore innocente* (*Innocent Pain*), Mancuso admits the former in order to ensure the latter. In brief, Mancuso argues that the world is governed by impersonal logic "which proceeds without regard for individuals", allowing various handicaps because "the personal God has willed it so, as the only indispensable condition for the birth of freedom' and also of love. Freedom, however, has a high price, since it 'consists in the very real possibility of evil, with all the burden of pain it brings".[28] In fact, as Mancuso writes, "under the fiery rays of divine necessity, the seedling of freedom would wither before it was even born". Mancuso, therefore, suggests that the personal God, God the Father, withdraws from Creation. He exists, yes, but he is absent here and now while at the same time present in a different way and elsewhere. God, Mancuso emphasizes using scientific-like phrasing, "is donation of energy, is energy that gives itself", and that energy is the source of the impersonal nature that leads to the emergence of freedom in man. To make this freedom possible, "God the Father creates man and by creating him gives him the Son through the action of the Spirit. In this sense we are created in the Son", and "everything was made through him" (Mancuso 2009, p. 157). In short, to account for the freedom of man, Mancuso proposes a doctrine of the creation

of the world by God the Father with a contextual handing over of the world to the Son. In this way, creation is marked by the Incarnation and, at the same time, handed over to impersonal forces that govern the evolution of the natural world.

The core of Mancuso's *Gestalt* is expressed in the following quote:

> The distinction between spirit and matter, as contemporary physics has shown, ultimately does not exist, being that all that exists is energy. The Big Bang is the irruption of God's energies in the physical world, the cross is the irruption of the Son of God in the human world: God crucified in nature, God crucified in history. The Big Bang was the rupture of the original symmetry that held the four fundamental forces together. From this rupture, this tearing of the original unity, comes the energy that continuously generates the cosmos. Creation, from the physical point of view, is the diffusion of primordial energy, its incessant transformation until the generation of intelligent and spiritual life, which represents the stage in which energy completes its cycle: spirit, pure energy, becomes matter that then becomes spirit again.
>
> (Mancuso 2009, p. 160)

In Mancuso's complex narration about the Creation, one can see how purely scientific language (energy/Big Bang/symmetry rupture) intertwines with exquisitely theological language (Creation/Incarnation/the Cross). The resulting narration gives meaning to the suffering of handicapped persons within the *Gestalt*, in which the fabric of the world is described in terms of energy, God's withdrawal, and Incarnation. However, human liberty is preserved, notwithstanding its price. It is even possible to identify in Mancuso's approach the abductive pattern of reasoning from the effect (suffering) to a cause (the maintenance of liberty), which underpins the entire argument, leading to the construction of the corresponding mental model.

Mancuso's *Gestalt* is highly believable and is distinguished by a significant degree of narrative fidelity. Using Fehige's wording (see Fehige 2019, p. 1), one can say that Mancuso tells how God runs the cosmos. His narration *is* a mental model. Its *aim* is to give sense to the suffering of the innocent. Its *topic* is men's freedom. In general, that of Mancuso is a *theology of science* narration.

However, subordinating theological discourse to a scientific one does not fully respect the Jewish corrective. This is particularly evident when Mancuso addresses the issue of verification principles. One of them requires the translatability of the basic concepts of the proposed *Gestalt* and, by and large, of theology into the language of science. Mancuso writes:

> God is the source of life, repeats the pontifical Magisterium, taking up the teaching of the Bible and Tradition. This is an affirmation that not only concerns spiritual life, but also life in its entirety, including the physical dimension. It is necessary for such an affirmation to be subjected to science in order to have a translation in terms of contemporary science if it is not an empty problem.
>
> (Mancuso 2009, p. 205)

In some of his writings, Mancuso goes as far as to support a thesis according to which the laws of nature are to be identified with the name of God, understood as the Ordering Principle that "injects into the cosmos the information necessary for the complex process of life to take place" (Mancuso 2013, p. 401). In this way, Mancuso delegates to science the role of the interpreter of the Bible, violating the principle of modesty (see P3 above).

## 5. Conclusions: Let Nature Speak—Theology of Science as a Possible Means of Narrative Consilience between Science and Theology in the Modern Age

Judith Oster recalls Eli Wiesel's assertion that "God created human beings because He loves stories—a conclusion Weisel reaches based on the Chasidic tale of the Baal Shem Tov" (Oster 1999). What kind of stories? The great Rabbi Israel Bal Shem Tov, as Wiesel recounts, was meditating on the misfortunes of the Jews and was accustomed to visiting a certain

place in the forest to light a fire and pray. And the hardships were alleviated. Years passed, and his disciple, the celebrated Magid of Mezritch, had to pray for the same reason, but he remembered only the place and the prayer, but not how to light the fire. Nevertheless, it was enough for the miracle. Again, years passed and the same problems returned. So, Rabbi Moshe-Leib of Sasov, to save his people, went to the forest, but he remembered only the place and not the prayer nor how to light the fire. Even in that case, it was enough for the miracle. When many years later, in the same troublesome situation, it was the turn of Rabbi Israel of Rizhyn. He sat in his armchair and spoke to God: "I am unable to light the fire and I do not know the prayer; I cannot even find the place in the forest. All I can do is to tell the story, and this must be sufficient. And it was sufficient" (Oster 1999).

Here is the forest and the fire (the natural world), here is the prayer to God for his chosen people (the Bible), and here are rabbis who combine these elements to solve an existential problem of the Jews. In the end, after many, many years, when the link between the forest, the fire, and the prayer was forgotten, there remained only the narration which unites all these elements. As far as the forest, the prayer, the rabbis, and God are considered, Wiesel's story can be interpreted, at least to some extent, as a metaphor for genesis and a need for theology of science narration. It is a story about how the connection between the natural world and the teachings of the Bible can easily be forgotten. It is important to tell this story, especially when considering that narration is especially needed in difficult times.

What about the present time? In a brilliant synthesis, Balthasar enumerates three stages of the road that led Western spiritual history to where it is now:

> In a certain sense three phases can be distinguished: a first in which the ancient solutions retain a strong poignancy as a form of thought and expression within the Christian message; a second in which, due to the fading of the Christian message, links to ancient motifs re-emerge more uncovered; and finally a third in which, following the radical changes leading to the modern vision of the world, the ancient forms also fade away, freeing up space for the new.
>
> (Balthasar 2019, p. 17; see also Lococo 2002, p. 12)

'The new' most likely stands for the vision and perception of the world dominated by science and technology. What does it mean? In the essay *The Question of God and Modern Man*, which appears to be Balthasar's only organic writing on science and the scientific worldview, the Swiss theologian stated: "if nature is understood as what it is meant to be for man, it can and must speak to us with the language God has given it. The question, however, is whether we do not prevent it from doing so" (Balthasar 1967, pp. 25–26).

It seems that as far as the Bible and the Book of Nature are considered, humankind is in the situation of Rabbi Israel of Rizhyn, which is sufficient reason to search for the story. But is the story necessary only in difficult times? Or rather, is it true that times are always both good and bad? Consider Charles Dickens's memorable incipit: "It was the best of times, it was the worst of times, it was the age of wisdom, it was the age of foolishness, it was the epoch of belief, it was the epoch of incredulity, it was the season of Light, it was the season of Darkness, it was the spring of hope, it was the winter of despair" (Dickens 1981, p. 1). If this is the case—if times are always good and bad, and thus there is no choice—the story must be told constantly. This observation demonstrates the necessity of theology of science.

In short, theology of science is essentially a narrative enterprise. Theology of science narration does not lead to anything that is solely scientific, nor to anything that is solely theological. Instead, theology of science lights up sparkles of meaning in the natural world when the latter is illuminated by a supernatural ray because, following Józef Tischner's intuition, 'meaning' is a modern name for 'light' (Tischner 2020, p. 164). Furthermore, "the light that causes everything to emerge in such a way that it is evident and comprehensible in itself is the light of the word" (Gadamer 2006, p. 478). Therefore, theology of science, thanks to its storytelling ability and creativity, is able to "[decipher] the world, which without that narrative would remain unknown and mute", to use Fisher's words in reference to the rhetorical art of Ernesto Grassi (Fisher 1985, p. 11). One theology of science narrative

engenders another; this is due to the intricate intertwining of the inexhaustible depth of significance of the Bible and the ever-expanding reading of the Book of Nature. While the thematic connection 'light-meaning-word-narrative' in the context of theology of science requires further exploration, it is exactly at this intersection that the essentiality, necessity, and irreplaceability of theology of science are rooted. It is, therefore, possible to claim that theology of science narration makes divine reality shine in nature, and permits the Book of Nature to "speak to us with the language God has given it" (Balthasar 1967, p. 26).

**Funding:** This research received no external funding.

**Data Availability Statement:** No new data were created or analyzed in this study. Data sharing is not applicable to this article.

**Conflicts of Interest:** The author declares no conflicts of interest.

## Notes

1. I have written 'Christian-Catholic theology', because one of the basic concepts, namely the specific domain of theology of science, stems from the insights of Hans Urs von Balthasar, an eminent Catholic theologian. However, the concept of theology of science described in this essay could also be applied to non-Catholic Christian theological currents, as it is based on the idea of an intertextual reading of the two Books. Further studies are required to undertake such an extension, which was not possible here due to space constraints.

2. (Agazzi 2014, p. 2). Agazzi's approach seems to align with the following statements of Balthasar: "certainly in the first rush of victory science did indeed try to apply to every sphere of life its own simplest and seemingly safest method, that is the principles of mathematics, physics, and chemistry. Yet this procedure contradicted its own program because it was not objective and failed to take into account the special claims of the object. Today this is being recognized, at least in the Western world. It was the principal result of the phenomenological movement to clarify these methodical questions of science and scholarship and *to demand for each sphere of knowledge its own appropriate method.* The deeper layers of nature will not answer to a superficial questioning, one that is suitable only for the lowest sphere of matter" (Balthasar 1967, p. 25; my italics to underline the analogy between the basic ideas of Agazzi's and Balthasar's synthesis). For a similar approach, see (Torrance 1969, p. 341).

3. In some way, Agazzi's theory of 'objectivity' is an answer to what Lococo describes as "an 'objectivity' fetish", a consequence of "a superficial appropriation of the method proper to natural science and a misunderstanding of the nature of theology". This misunderstanding, in Lococo's words, creates a situation in which "science views theology as a subjective world view that has nothing to do with 'real' knowledge" (Lococo 2002, p. 11). As to the objectivity in theology, see Karl Barth's considerations on the knowability of the word of God in (Barth 2010a, pp. 184–244).

4. In his essay *The God Question and Modern Man*, Balthasar rephrases this idea as follows: "the two realities of science and Christianity, though seemingly unconnected, are nevertheless related by an intermediate sphere. From the scholar's point of view this appears as *Weltanschauung*, from the Christian's as 'religion', and from the centre as philosophy" (Balthasar 1967, p. 10).

5. (Balthasar 2004, p. 51). Regarding the translation of *esse* in English, see, for example (LaZella 2010, pp. 4–5).

6. For the most recent history of this important letter, see (Camerota et al. 2019).

7. Dogmatic Constitution on Divine Revelation, solemnly promulgated by His Holiness Pope Paul VI, on 18 November 1965, *Dei Verbum* 4. Wherever I write about the Bible, I keep in mind what was stated in the present paragraph.

8. An inspiring analysis of David Brown indicates that the source of Balthasar's fallacy could be the equation of glory and beauty (for details, see Brown 2018).

9. In other terms: "what Physics describes is real/what the Bible says is real/and human's love and pain are real", as a paraphrased version of the poem *Veritas* by Halina Poświatowska, affirms ("fizyka jest prawdziwa/biblia jest prawdziwa/miłość jest prawdziwa/i prawdziwy jest ból"; Poświatowska 1989, p. 240).

10. Myrna Solotorevsky, in her essay on Borges and Midrash, writes: "the correspondences between Borges and Midrash [are] in the idea of intertextuality, in the concept of reading not as lineality but as a configuration of textual space, in the notion of destructuralization of the text as a condition for deciphering it" (Solotorevsky 1986, p. 263).

11. In this attempt at defining intertextuality and narration, as practised in theology of science, I draw upon Kate Finley's and Joshua W. Seachris's thinking (Finley and Seachris 2021).

12. As to the models of narrative theology, see, for example (Mauz 2009; Finley and Seachris 2021).

13. For a very similar approach, see (Elgin 2002). Catherin Elgin, in her book *True Enough*, notes that "disciplinary understanding is not an aggregation of separate, independently secured statements of fact; it is an integrated, systematically organized account of a domain" (Elgin 2017, p. 13).

14. See, for example, Jacques Hadamard's excellent description of this process in mathematics (Hadamard 1954; Sadler-Smith 2015) or Di Rocco's description with reference to John Dewey's and Arthur Bentley's theory of knowledge (Di Rocco 2018, pp. 107–28).

[15]    For an overview of recent developments, see (Johnson-Laird 2004; Stuart et al. 2018).

[16]    A "thought experimenting is a form of simulative model-based reasoning", writes Nancy Nersessian (1993, p. 291). See also (Fehige 2019, pp. 5–9).

[17]    (Huxley 1986, pp. 83–84). Likewise, Roman Ingarden notes in his work on the literary work of art that what he calls 'metaphysical qualities', like, for example, the sublime, the inexplicable, the tragic or the holy, "make life worth living" (Ingarden 1973, p. 291). I believe this is an appropriate place for a rather important digression. The analysis of the discourse of theology of science is conducted in this paper through the lens of a narrative paradigm. The reference to Ingarden wants to suggest that the concept of literary work of art developed by the Polish phenomenologist, for instance, may be more suitable for the analysis of a narrative of theology of science than the tools developed within the framework of philosophy of science. This subject matter would merit further investigation.

[18]    (Bruner 1991, p. 4). Regarding narrative reasoning and narrative time, see also (Patron 2006; Worth 2008; Ricoeur 1980).

[19]    For a synthesis, see (Stroud 2016, pp. 2–4).

[20]    (Worth 2008, p. 48), with reference to J. Bruner. The logico-scientific reasoning is sometimes called 'the clear-cut inferential' and is considered a standard account of argument (see Stroud 2016, p. 1).

[21]    The illative sense has an important place in Balthasar's synthesis; see (Balthasar 1990, p. 132).

[22]    In my synthesis, I rely on (Banon 2009; Stern 1998; Wiesel 1976); see also (Law-Vilijoen 1997; Taggart 2010; Carruthers et al. 2014).

[23]    Christopher B. Kaiser does not elaborate on what is referred to here as the $B^2N^2$ Model. However, in his stimulating book *Toward a Theology of Scientific Endeavour: The Descent of Science* (Kaiser 2007), he makes extensive use of the Midrash *Bereshit (Genesis) Rabbah* as a theological resource in Chapters 1 and 4.

[24]    Stern's reflections are similar to Erich Auerbach's considerations on the two modes of representing reality in European culture. Auerbach scrutinizes the Homeric style (Euryclea recognizes Odysseus), which is 'of background', and the Elohistic style (the account of the sacrifice of Isaac), which is 'fraught with background'. In *Mimesis*, he writes, "we have compared these two texts, and, with them, the two kinds of style they embody, in order to reach a starting point for an investigation into the literary representation of reality in European culture. The two styles, in their opposition, represent basic types: on the one hand fully externalized description, uniform illumination, uninterrupted connection, free expression, all events in the foreground, displaying unmistakable meanings, few elements of historical development and of psychological perspective; on the other hand, certain parts brought into high relief, others left obscure, abruptness, suggestive influence of the unexpressed, 'background' quality, multiplicity of meanings and the need for interpretation" (Auerbach 1953, p. 23).

[25]    On this issue, see also: (Schumann 2012, pp. 150–64; Tirosh-Samuelson 2018, p. 399).

[26]    See above, Section 2, and (Fehige 2019, p. 2).

[27]    (Mancuso 2009, all translations from Mancuso are mine). On Vito Mancuso as a lay theologian, see (Simuţ 2011a, 2011b).

[28]    See (Mancuso 2007, pp. 122 and 116). Mancuso's words are similar to Galilei's assertion from his *Letter to Benedetto Castelli*: "nature is inexorable and immutable, and she does not care at all whether or not her recondite reasons and modes of operations are revealed to human understanding" (Galilei 1613).

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
