# Peer review of "Theology of Science as an Intertextual Reading: The Bible, the Book of Nature, and Narrative Paradigm"

_religions, doi:10.3390/rel15030293_

Round 1

Reviewer 1 Report

Comments and Suggestions for Authors

Theology of science is an area that has received some attention of late--e.g. Robert Trundle (2007, Michael Heller (2008), Catherine Punsalan-Manlimos (2010), and Tom McLeish (2014) to name few. As McLeish has pointed out, because purse scientific pursuit tends to undermine all that is most precious to humanity, a "theology of science" can bridge this divide by assuming a biblical mandate to investigate that natural world. In this way our scientific pursuits can--and indeed are meant--to be part of God's plan. Here, unlike natural theology, scriptures is enlisted on the behalf of scientific inquiry. 

I am sympathetic to such a view and the present article seeks to further the idea of a "theology of science" by bringing in the ideas of Hans Urs von Balthasar. Following modern philosophy's linguistic bent, it seeks to build this upon what it calls "the narrative paradigm." Perhaps, but I'm not entirely convinced that this so-called "theology of science" brings much new to what might otherwise be call natural theology. One is reminded of Christ's caution against putting new wine (theology of science) into old wine skins (natural theology) in Luke 5:37-39.

My skepticism notwithstanding, this new approach to conjoining science and theology should have its say. While the argument is intriguing, I am thoroughly unimpressed with the author's reliance upon concilience as, citing Larry Lauden's words, "vital to a comprehension of the philosophy of science" and the "key to the historiography of science." If concilience is meant to say that the strength/validity of a theory is made more so (confirmed even) by the mere fact of its explanatory power across fields of inquiry and disciplines, then history is full of counter example. (If we mean something else by concilience, a better term should be adopted.) In any case, phlogiston, once thought to explain a great many chemical and physical reactions, was utterly false. The humor pathology of Galen (imbalances of blood, phlegm, yellow bile and back bile) was once thought to explain all manner of acute and chronic diseases, even mental disorders. We know the "concilience" of humor pathology to be one of great red herrings of medical science. 

Having said all this, I am not willing to cast aside the concept of a theology of science altogether, as Philip Kitcher and Lewis Walpert (for difference reasons) has done. Still, it is a niche concept popular in modern Catholicism yet to take hold. For that reason, more discussion along these lines should be encouraged. In short, despite my reservations, it should be published.

Comments on the Quality of English Language

The language and writing still of this essay needs help. The frequent use of Latin phrases and terms give the piece a pedantic, inkhornish quality. By inkhornish I refer to Jeffrey Kacirk's usage of the word as "overworked and unnecessarily intellectual, or inkhornish, perhaps from too much burning of the midnight oil." Such writing is meant more to impress than explain; it ends up doing neither. My only call for revision is to write more clearly so that it can be equally understood by theologians, philosophers, and scientists. Otherwise, whatever the merits of this essay, it will be little read and soon forgotten. In short, improve the "narrative" first, let the "paradigm" take care of itself.

Author Response

Dear Sir,

Thank you very much for your review, very stimulating and accurate. In response to your comments, I would like to say what follows.

1). As for the possibility of extending the approach to non-Catholic Christian theologies I am generally of the opinion, that the approach presented in this essay is applicable to all theologies. I have written in my text ‘Christian-Catholic theology’, because one of the basic concepts, namely the specific domain of theology of science, stems from the insights of Hans Urs von Balthasar, an eminent Catholic theologian. However, the concept of theology of science described in this essay could also be applied to non-Catholic Christian theological currents, as it is based on the idea of an intertextual reading. Further studies are required to undertake such an extension, which was not possible here due to space constraints.

2). As far as the originality of my approach is concerned - this is clearly outlined in the conclusion of the Introduction, where I wrote that 'if there is something original in the present paper, it is the novel arrangement of ideas previously expressed by various thinkers, theologians, philosophers, and scientists' - then it is a novelty in the sense of Catherine Elgin.

Furthermore, I would like to emphasise, my approach has nothing to do with natural theology. It is based on an intertextual reading of the two books, and is in no way intended to deal with the existence or characteristics of God on the basis of natural facts.

3). I would also like to clarify that my reference to Larry Laudan is not to his concept of scientific research programmes, nor to any of his metascientific concepts. I am only referring to Laudan's commentary on Whewell's philosophy, so - if anything - it is a reference to Whewell via Laudan, and not to Laudan himself.

4). Finally, I have tried to look for a remedy to what you have called 'inkhomish' aspects of my language. Thank you very much for that observation.

Thank you once again, and thank you for the time you have dedicated to the paper.

Best regards.

Reviewer 2 Report

Comments and Suggestions for Authors

132-133, 136: the real distinction between essence and reality (essentia-esse): I do not think that in this context the rality is a good translation of esse...

754 remove indentation, it's not a quote

Author Response

Dear Sir,

Thank you very much for your review, very accurate. 

I have correceted what you have indicated in your review. to the paper.

The main difficulty was with the term "esse". I have tried in that way: 

"Perhaps the most relevant element of this domain from Balthasar’s point of view is the real distinction between essence and being (essentia-esse), which leads to the contingency of the world, that ...". To pojnt is that Balthasar writes: "the coinciding of essence and reality (essentia-esse), ...". But I think you are rigth - so in my comment I use "being", in Balthasar quote - as above. 

Thank you once again, and thank you for the time you have dedicated to the paper.

Best regards.